# Tumor Characteristics Associated with Axillary Nodal Positivity in Triple Negative Breast Cancer

**DOI:** 10.3390/diseases11030118

**Published:** 2023-09-08

**Authors:** Neha Chintapally, Katherine Englander, Julia Gallagher, Kelly Elleson, Weihong Sun, Junmin Whiting, Christine Laronga, Marie Catherine Lee

**Affiliations:** 1University of South Florida Morsani College of Medicine, Tampa, FL 33602, USA; nehac@usf.edu (N.C.); englanderk@usf.edu (K.E.); gallagher20@usf.edu (J.G.); 2Regional Breast Care, Genesis Care Network, 8931 Colonial Center Dr #301, Fort Myers, FL 33905, USA; kelly.elleson@moffitt.org; 3Comprehensive Breast Program, Moffitt Cancer Center, Tampa, FL 33612, USA; unmin.sun@moffitt.org (W.S.); unminne.laronga@moffitt.org (C.L.); 4Department of Biostatistics & Bioinformatics, Moffitt Cancer Center, Tampa, FL 33612, USA; unmin.whiting@moffitt.org

**Keywords:** breast cancer, axillary lymph node metastasis, triple negative subtype

## Abstract

Larger-size primary tumors are correlated with axillary metastases and worse outcomes. We evaluated the relationships among tumor size, location, and distance to nipple relative to axillary node metastases in triple-negative breast cancer (TNBC) patients, as well as the predictive capacity of imaging. We conducted a single-institution, retrospective chart review of stage I–III TNBC patients diagnosed from 1998 to 2019 who underwent upfront surgery. Seventy-three patients had a mean tumor size of 20 mm (range 1–53 mm). All patients were clinically node negative. Thirty-two patients were sentinel lymph node positive, of whom 25 underwent axillary lymph node dissection. Larger tumor size was associated with positive nodes (*p* < 0.001): the mean tumor size was 14.30 mm in node negative patients and 27.31 mm in node positive patients. Tumor to nipple distance was shorter in node positive patients (51.0 mm) vs. node negative patients (73.3 mm) (*p* = 0.005). The presence of LVI was associated with nodal positivity (*p* < 0.001). Tumor quadrant was not associated with nodal metastasis. Ultrasound yielded the largest number of suspicious findings (21/49), with sensitivity of 0.25 and specificity of 0.40. On univariate analysis, age younger than 60 at diagnosis was also associated with nodal positivity (*p* < 0.002). Comparative analyses with other subtypes may identify biologic determinants.

## 1. Introduction

Breast cancer remains the most common cancer among women worldwide and the second most common cancer overall [1]. Approximately one in eight women will develop breast cancer in their lifetime, translating into around 300,000 new cases of breast cancer and 43,700 breast cancer-related deaths estimated in the United States in 2023 [2].

Triple-negative breast cancer (TNBC) is a distinct subtype of breast cancer characterized by the absence of estrogen receptors (Ers), progesterone receptors (PRs), and human epidermal growth factor receptor 2 (HER2/neu) cell surface proteins. TNBC accounts for approximately 10–15% of all invasive breast cancers and tends to have higher histological grades and an increased risk of metastasis [3,4]. The risk of TNBC is higher in women younger than 40, Hispanic and African-American women, and individuals with BRCA1 and PALB2 genetic mutations [5,6].

The current five-year relative survival rate for TNBC is 77% compared to 90% for all breast cancer subtypes combined [3,7,8], with metastatic spread being the leading cause of death [9]. Due to their lack of ER, PR, and HER2 cell surface proteins, TNBC cells do not respond to targeted treatments commonly used for other subtypes. Surgical management of TNBC includes upfront resection through lumpectomy or mastectomy, followed by radiation and/or chemotherapy [10,11]. However, before initiating any treatment, the assessment of axillary lymph node involvement is essential.

TNBC patients frequently present with nodal metastases at the time of diagnosis [12,13], which portends a worse survival outcome compared to those without nodal involvement [14]. Sentinel lymph node biopsy (SLNB) is the primary technique used to identify nodal metastases in breast cancer. Various tumor characteristics can aid in predicting nodal positivity, including tumor size, tumor location, and the distance of the tumor from the nipple. Larger tumor size at diagnosis and shorter tumor to nipple distance have been associated with an increased likelihood of nodal involvement [15,16,17].

In the past, axillary staging was performed with the highly morbid axillary dissection procedure. Today, patients who are considered clinically node negative typically undergo a sentinel lymph node biopsy (SLNB) as the initial approach [18]. The potential complications of SLNB include seroma formation, lymphedema, and pain. Though approaches have been explored to reduce complications, such as the use of fibrin glue during SLNB or ALND, studies have shown mixed results regarding their efficacy in preventing seromas [19]. These complications highlight a need for alternative approaches to axillary staging. Given the concurrent advances in imaging technology, preoperative radiographic imaging can be utilized to assess nodal involvement before proceeding with surgical management, with a goal of minimizing the invasiveness of SLNB and improving patient outcomes [20]. Preoperative axillary ultrasound, however, has lower sensitivity than SLNB, with positive predictive value of 76.5% and negative predictive value of 51.9% [21].

Our study aims to evaluate the relationship between tumor characteristics (tumor size, tumor to nipple distance, and tumor location relative to lymph node positivity) and axillary lymph node involvement, specifically in the TNBC subtype. Additionally, the study’s secondary objective is to assess the capacity of radiographic imaging, particularly axillary ultrasound, in identifying suspicious axillary lymph nodes. Enhancing the accuracy of preoperative imaging techniques could potentially reduce the need for invasive procedures, such as SLNB, and improve the overall patient experience. By examining the correlation between radiographic findings and nodal involvement, we can determine the utility and limitations of axillary ultrasound as a diagnostic tool for TNBC patients.

## 2. Materials and Methods

This study was a single-institution, IRB-approved, retrospective review of female breast cancer patients. As depicted in Figure 1, data were gathered from medical records, billing records, and pharmacy records from 1 January 1998 through 31 January 2019. Preliminary data collection included patients who had been diagnosed with ER+, PR+, and HER2/neu, or TNBC, and who underwent upfront surgical tumor resection at a comprehensive cancer center. From this study, we only included patients who had been diagnosed with stages I–III TNBC and underwent upfront surgical tumor resection at a comprehensive cancer center. TNBC patients who presented with recurrent, metastatic, inflammatory, bilateral, or multicentric primary tumors were excluded from this study.

Clinical, imaging, and pathologic data were collected, including tumor size, tumor distance from the nipple, location of the tumor, and the imaging modality used to visualize lymph nodes. Demographics, last follow-up date, and status were also collected. Tumor size was collected as a continuous variable and recorded in millimeters (mm). Tumor location was categorized according to quadrant. Using the clock-face method, tumors were categorized as upper inner quadrant (UIQ), upper outer quadrant (UOQ), lower inner quadrant (LIQ), or lower outer quadrant (LOQ). Tumors were classified as overlapping if they were reported to be located at the 12 o’clock, 3 o’clock, 6 o’clock, or 9 o’clock positions. Follow-up status was classified as alive without disease, alive with disease, or deceased. Imaging data were collected from MRI, mammogram, and ultrasound reports. We reviewed radiologists’ impressions of radiographs to determine the nodal findings for each patient. ANOVA or the t test was used to evaluate the association of continuous variables and lymph node involvement. The chi-square test or Fisher’s exact test if applicable was utilized to evaluate the association of categorical variables with lymph node positivity. A univariate logistic regression model was applied to determine the impact of each predictive variable on lymph node positivity. A multivariate logistic regression model was applied to determine the combined impact of multiple predictive variables on lymph node positivity. All statistical tests performed were two-sided, with the level of significance established at *p* < 0.05. All analyses were performed using SAS software (version 9.4, SAS Institute Inc., Cary, NC, USA).

## 3. Results

### 3.1. Demographic Information

A total of 728 female patients diagnosed with primary invasive breast cancer between 1998 and 2019, aged 21 to 96 years old (median age 61 years) and treated with primary resection, were identified in our review from a single institution’s prospective database. Of the total, 400 women (54.9%) underwent breast conservation surgery, and 328 women (45.1%) underwent mastectomy. A majority (40.8%) of tumors were T1c and invasive ductal carcinoma (84.1%), with a histologic grade of 2 (49.9%). Most tumors did not show lymphovascular invasion (81%) and were ER positive (86.1%) and PR positive (80.2%), without HER2 overexpression (88.5%) The median follow-up period was 40.6 months (ranging from 0.3 to 242 months). All 728 patients were clinically node negative, and 99.9% of women underwent an SLNB. Among them, 351 patients (48.2%) had at least one positive lymph node, of whom 231 patients underwent an ALND. The median number of nodes obtained during ALND was 15 (ranging from 1 to 43).

From this larger dataset of 728 patients, we identified 73 patients with TNBC. As shown in Table 1, 73 patients diagnosed with primary TNBC from 1998 to 2019 had a mean age of 56.3 years old (range 21–85 years). Figure 2 shows the distribution of tumor characteristics among the TNBC cohort. The T stage distribution was T1mi (2.7%), T1a (11%), T1b (4.1%), T1c (38.4%), T2 (42.5%), and T3 (1.4%). Of the patients, 91.8% had grade 3 tumors. The mean tumor size was 20 mm (range 1–53 mm), and the mean follow-up time was 48 months (range 5–142 months). Thirty-six patients underwent a lumpectomy, and 37 patients underwent a mastectomy. Of the patients, 17.8% had recurrence of TNBC, of whom 7 had distant recurrence, 2 had local recurrence, and 4 had locoregional recurrence. Of the patients, 13.7% died due to breast cancer-related complications by the end of the follow up period. All patients were clinically node negative, with 43.8% being sentinel lymph node positive. Twenty-five patients underwent ALND. Of the patients, 25.9% had lymphovascular invasion.

### 3.2. Analysis of Tumor Size and Nodal Positivity

Table 2 highlights the associations found between tumor characteristics and axillary node status. Thirty-two of 73 (43.8%) of the TNBC patients identified were found to be positive for sentinel lymph node involvement; six (8.2%) TNBC patients were identified as having sentinel lymph node (SLN) and further axillary nodal metastasis. Increasing primary tumor size was significantly associated with SLN positivity (*p* < 0.001); the average tumor size for node negative patients was 14.3 mm compared to 27.3 mm for node positive patients. Larger tumor size was also significantly associated with total node positivity (*p* < 0.001). We noted a significant association between SLN positivity and T stage (*p* < 0.001), with 23 (74.2%) SLN positive patients at T stage 2. One (3.1%) SLN positive patient was noted to be at T stage 3. Patients who underwent ALND were more likely to have a larger tumor size (*p* < 0.001). The average tumor size for patients who underwent dissection was 30.2 mm compared to 14.6 mm for patients who did not undergo the procedure (*p* < 0.001). We also noted that larger primary tumor size was associated with higher rates of lymphovascular invasion (*p* = 0.014) and a greater likelihood of disease recurrence (*p* < 0.001). The average tumor size was 31.7 mm for the 13 cases with disease recurrence versus 17.5 mm for the 60 cases without recurrence (*p* < 0.001). A similar association was noted between T stage and disease recurrence (*p* = 0.015).

### 3.3. Analysis of Lymphovascular Invasion, Tumor Location, Distance to Nipple, and Nodal Positivity

The mean distance to the nipple as detected by ultrasound was 65.5 mm (range 10–140 mm). Shorter distance to the nipple was significantly associated with an increased likelihood of axillary nodal involvement (*p* = 0.005). The average distance to the nipple was 51.0 mm for SLN positive patients compared to 73.3 mm for node negative patients. Thirty-three patients had a tumor located in the axillary tail/upper outer quadrant (UOQ), nine patients had a tumor located in the upper inner quadrant (UIQ), seven patients had a tumor located in the lower inner quadrant (LIQ), 12 patients had a tumor located in the lower outer quadrant (LOQ), and 12 patients had a tumor located in overlapping regions. No significant association was found between tumor location and nodal positivity (*p* = 0.653). For this cohort, the presence of lymphovascular invasion was significantly associated with nodal positivity (*p* < 0.001).

### 3.4. Predictive Capacity of Imaging in Detecting Nodal Positivity

Twenty-one patients had an axillary ultrasound (AUS), 19 patients had an MRI, and 10 patients had a mammogram performed to detect suspicious nodes preoperatively. The sensitivity and specificity of imaging for nodal metastasis were 0.19 and 0.67, respectively. Preoperative imaging detected abnormal nodes in 49 patients. AUS yielded the largest number of suspicious findings (21/49) and had sensitivity of 0.25 and specificity of 0.40. We were unable to draw conclusions regarding the associations of tumor size, T stage, tumor to nipple distance, SLN size, and number of positive nodes with the likelihood of suspicious nodal findings on imaging.

### 3.5. Univariate and Multivariate Regression Analyses of Factors Associated with Nodal Positivity

On univariate logistic regression analysis, age younger than 60 at time of diagnosis (*p* < 0.002), T stage (*p* < 0.001), larger tumor size (*p* < 0.001), and shorter tumor to nipple distance (*p* = 0.009) were all associated with ALN positivity (Table 3). On multivariate logistic regression analysis, larger tumor size (*p*< 0.001) and type of surgical procedure, particularly mastectomy (*p* < 0.001), were associated with nodal positivity (Table 4).

## 4. Discussion and Conclusions

TNBC is generally associated with higher grades and poorer prognoses; these tumors lack the cell surface proteins needed for targeted therapy. We sought to evaluate the relationships between axillary nodal metastasis and specific tumor characteristics, such as tumor size, distance to the nipple, location, and imaging results, for the subtype of TNBC. Axillary nodal metastasis is an important negative prognostic marker that guides the surgical management of primary TNBC and subsequent treatments. The results of our study showed that 25.9% of stage I–III TNBC patients had lymphovascular invasion, and 43.8% of patients had lymph node metastasis. In line with this finding, studies have suggested that TNBC patients have higher rates of lymph node metastasis (30–50% or higher) compared to the other subtypes [9,22,23]. The high rates of axillary metastasis in TNBC may be explained by alterations in gene expression. Multiple studies found that metastatic triple negative breast cancer cells had altered expression of various miRNAs involved in survival, proliferation, and genomic stability [9]. Another study cited the upregulation of chemotaxis and antiapoptotic genes and the downregulation of genes involved in maintaining the tumor microenvironment [22]. Four TNBC-specific genes, (ankyrin repeat domain 30A, acidic nuclear phosphoprotein 32 family member E, desmocollin-2, interleukin 6 cytokine family signal transducer) associated with enhanced survival and spread have also been identified [9]. Notably, other breast cancer subtypes lack the expression of these genes, which may provide pathological insight for increased rates of axillary metastasis in TNBC patients.

We identified two key independent predictors of axillary lymph node metastasis in patients with TNBC. The first independent predictor of axillary metastasis is primary tumor size in millimeters. This finding is consistent with various studies that have explored the implications of increasing primary tumor size in all subtypes [24,25]. Our analysis also showed an increased risk of cancer recurrence with larger tumor size and increasing T stage. Similar findings were described in a study by Lafourcade et al., who noted that higher grade and larger tumors were associated with a higher risk of recurrence and death [26].

The second predictor that we identified was tumor to nipple distance. Our analysis showed that decreasing tumor to nipple distance was associated with an increased likelihood of axillary nodal involvement in TNBC patients. Thus far, there have been conflicting reports detailing the role of tumor to nipple distance in axillary metastasis [17,27,28]. The pathogenesis of metastasis by tumors located closer to the nipple may be explained by the proximity to the nipple–areolar complex and the anatomy of the breast lymphatic drainage system, which facilitates chemotaxis and spread, although this theory has not yet been well researched or substantiated [28]. Supporting this hypothesis, studies have identified that tumors located in the retro-areolar region of the breast, where the main lymphatic vessels pass, have higher rates of axillary metastases [25]. A study by Yoshihara et al. noted that both retro-areolar and lateral breast tumors are more likely to metastasize to the axilla [25,29,30]. A study by Xiong et al. drew associations between tumor location in the UOQ and axillary metastasis [31]. We noted that a majority (33/73) of primary tumors were located in the UOQ/axillary tail, of which 15 involved axillary metastases. However, our analysis did not identify a significant association between tumor location and axillary metastasis. One potential explanation for this discrepancy may be that none of the TNBC tumors in our subset were reported to be located in the retro-areolar region; this fact is likely one of the significant shortcomings of retrospective studies. Further research is needed to delineate the importance of tumor location in predicting nodal disease.

Preoperative imaging can be an important diagnostic tool in breast cancer patients to assess axillary involvement and potentially replace the more invasive and risky procedures, such as SLNB and ALND [32]. In some institutions, evaluation of abnormal lymph nodes includes core needle biopsy or ultrasound guided fine needle aspiration (US-FNA). These techniques are often performed after evidence of nodal disease is obtained clinically through physical examination or imaging [33]. However, per the guidelines of Z0011, US-FNA is an unreliable method for determining which patients require ALND, resulting in overtreating 43% of patients with positive results and potentially denying necessary treatment to a noteworthy proportion of patients with negative results [33,34]. Hence, it becomes increasingly important to assess the role of imaging modalities, such as ultrasound, in identifying nodal disease. We particularly assessed the abilities of ultrasound, MRI, and mammogram to detect suspicious mammillary and axillary lymph nodes in patients with TNBC. From imaging reports, lymph nodes were described as suspicious based on cortical thickness and echo textures. A majority of TNBC patients in our study underwent preoperative imaging using axillary ultrasound. Our analysis of the imaging modalities noted that ultrasound had the largest number of suspicious imaging findings (21/49), with sensitivity of 0.25 and specificity of 0.40. Another study found the sensitivity and specificity of ultrasound to be 0.83 and 0.62, respectively, in the preoperative evaluation of axillary lymph nodes [35]. These discrepancies in sensitivity and specificity may be attributed to inter-observer variability during imaging analysis. We found the sensitivity and specificity for all imaging modalities combined to be 0.19 and 0.67, respectively. These values are significantly lower than the sensitivity (0.90) and specificity (0.96) of SLNB in determining lymph node positivity [36]. Our analysis showed that none of the tumor variables (tumor size, T stage, distance to nipple, SLN size, number of positive lymph nodes) were statistically significant in increasing the likelihood of suspicious imaging reports. Imaging modalities, such as ultrasound, MRI, and mammogram, are not powerful enough as standalone tools in capturing lymph node positivity in TNBC patients. Based on our results and those of other studies, findings from preoperative MRI, ultrasound, and mammograms may be pooled and used adjunctively to SLNB in TNBC patients [36]. Another avenue currently being explored includes the use of wire-guided localization and intraoperative ultrasound for the detection of abnormal axillary lymph nodes in breast cancer patients. While this technique is the gold-standard for the detection of non-palpable breast lesions, few studies have explored its role in SLN detection [37].

Our study does have certain important limitations. First, the sample size of TNBC patients, at 73, is relatively small, and there is a high preponderance of small tumors due to the increasing use of neoadjuvant chemotherapy over the 20-year time frame during which these patients were identified. The small number of TNBC patients identified despite the lengthy time frame may be attributed to TNBC being a rarer subtype and that, over the time course of this study, an increasing number of patients were recommended to undergo chemotherapy prior to surgery and were thus eliminated from the study population. The reliability of our study may be improved by increasing the sample size. Since we performed a retrospective study, many of the variables that we evaluated lacked data points reported in patient charts, particularly data about tumor grade and suspicious findings on imaging. There is also a limitation in the ability to analyze long-term follow-up data due to the intrinsic retrospective design of the study. Last, all patients in our study were clinically node negative, meaning that diseased lymph nodes were not clinically identifiable in any of the TNBC patients. As such, the associations drawn between tumor characteristics and lymph node positivity are applicable to only clinically node negative TNBC. Further research is needed to draw conclusions for clinically node positive TNBC cohorts.

In conclusion, our study highlighted two important independent predictors of axillary nodal metastasis in patients with clinically node negative stage I-III TNBC. First, larger size primary TNBC tumors, and consequently higher T stage, are more likely to metastasize to the axillary nodal system. Second, primary TNBC tumors located closer to the nipple are also more likely spread to the axilla. We did not find any significant associations between tumor location by quadrant and axillary nodal involvement. Additionally, we found no associations among tumor size, T stage, tumor to nipple distance, lymph node positivity, and the likelihood of suspicious findings on preoperative imaging, including ultrasound, MRI, and mammography.

## 5. Clinical Implications and Future Directions

The findings of our research advance our comprehension of the predictive factors for nodal positivity in TNBC, offering valuable insights to guide surgical staging and patient management strategies. Our findings underscore that relying solely on imaging, especially axillary ultrasound, may not suffice as an accurate predictor of nodal positivity. However, the results of this study illuminate key variables such as age younger than 60 at diagnosis, increasing tumor size, advanced T stage, and shorter tumor-to-nipple distance, which collectively augment the likelihood of axillary nodal positivity in clinically node-negative TNBC patients. These findings open up new avenues for further investigation with larger TNBC cohorts, including clinically node-positive cases, with the potential to develop a comprehensive risk assessment model. Such a model could play a pivotal role in discerning the necessity of SLNB or ALND versus mastectomy and lumpectomy procedures while avoiding potential procedure-related complications and ensuring positive clinical outcomes for patients.

In addition to surgical management, novel approaches, such as immunotherapy and targeted therapies exploiting vulnerabilities specific to TNBC cells, should be explored. Also, addressing the disparities associated with TNBC diagnosis and treatment is of paramount importance. The higher prevalence of TNBC among certain populations, such as women of Hispanic heritage and African ancestry, highlights the need for equitable access to screening programs, genetic counseling, and comprehensive healthcare services. Collaborative efforts among clinicians, researchers, policymakers, and patient advocates should be fostered to develop innovative strategies for prevention, early intervention, and improved patient outcomes. Furthermore, investigating TNBC within the broader context of cancer has the potential to reveal common risk factors, therapeutic strategies, and, importantly, shared factors for predicting cancer metastases that may be applicable to other cancer types. By exploring the complexities of TNBC within the even larger framework of disease, we can take a crucial step toward improving diagnostic accuracy, refining treatment strategies, and ultimately enhancing patient outcomes.

## Figures and Tables

**Figure 1 diseases-11-00118-f001:**
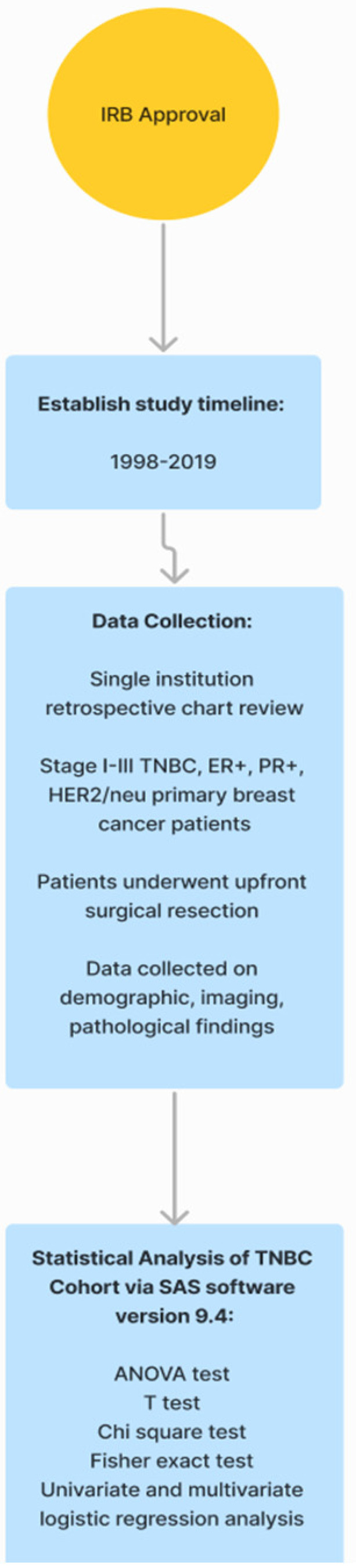
Strategic plan of the study. **TNBC:** triple-negative breast cancer, **ER+:** estrogen receptor positive, **PR+:** progesterone receptor positive, **HER2:** human epidermal receptor growth factor 2.

**Figure 2 diseases-11-00118-f002:**
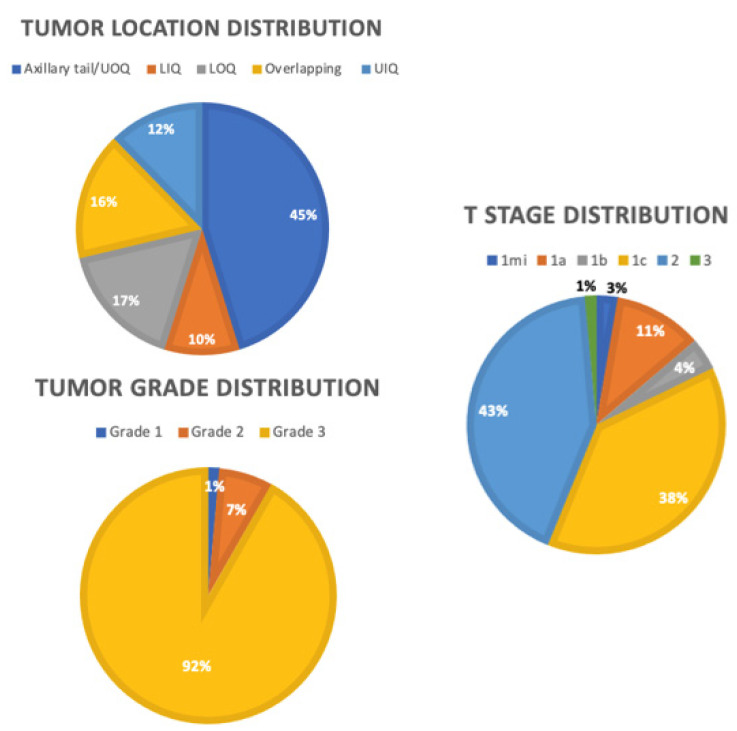
Distributions of tumor characteristics. **OUQ**: upper outer quadrant, **UIQ**: upper inner quadrant, **LOQ**: lower outer quadrant, **LIQ**: lower inner quadrant.

**Table 1 diseases-11-00118-t001:** TNBC Patient Demographics.

Variable	Level	*n* = 73	%
**Axillary lymph node** **dissection**	No	48	65.8
	Yes	25	34.2
**Grade**	1	1	1.4
	2	5	6.8
	3	67	91.8
**Alive status**	Alive	63	86.3
	Death	10	13.7
**Recurrence**	No	60	82.2
	Yes	13	17.8
**Recurrence type**	Distant	7	53.8
	Local	2	15.4
	Locoregional	4	30.8
**Sentinel Lymph Node**	Yes	73	100.0
**Stage T**	1mi	2	2.7
	1a	8	11.0
	1b	3	4.1
	1c	28	38.4
	2	31	42.5
	3	1	1.4
**Clinical node status**	Negative	73	100.0
	Positive	0	0.0
**Image type**	Mammogram	10	20.0
	MRI	19	38.0
	US	21	42.0
	Not applicable	33	-
**Tumor location**	Axillary tail/UOQ	33	45.2
	LIQ	7	9.6
	LOQ	12	16.4
	Overlapping	12	16.4
	UIQ	9	12.3
**Age at diagnosis**	Mean	56.3	
	Minimum	21	
	Maximum	85	
	Std Dev	13.0	
**Grade size (mm)**	Mean	20.0	
	Minimum	1	
	Maximum	53	
	Std Dev	11.4	
**Number of SLN positive**	Mean	0.5	
	Minimum	0	
	Maximum	3	
	Std Dev	0.6	
**Total number of suspicious nodes in image**	Mean	0.5	
	Minimum	0	
	Maximum	4	
	Std Dev	1.0	
**Tumor US distance to nipple (mm)**	Mean	65.5	
	Minimum	10	
	Maximum	140	
	Std Dev	31.0	
	Missing	10	
**Follow-up (months)**	Mean	48.3	
	Minimum	4.7	
	Maximum	142	
	Std Dev	30.6	

**SLN**: sentinel lymph node, **US**: ultrasound, **OUQ**: upper outer quadrant, **UIQ**: upper inner quadrant, **LOQ**: lower outer quadrant, **LIQ**: lower inner quadrant.

**Table 2 diseases-11-00118-t002:** Tumor Characteristics and Nodal Involvement.

Tumor Characteristics	Node Negative	Node Positive	*p* Value
	(*n* = 41)	(*n* = 32)	
**Mean Tumor Size (mm)**	14.3	27.3	***p* < 0.001**
**Mean Tumor to Nipple Distance (mm)**	73.3	51.0	***p* = 0.005**
**T-Stage**	2	0	***p* < 0.001**
1mi	6	2	
1a	2	1	
1b	23	5	
1c	8	23	
2	0	1	
3			
**Tumor Quadrant**	18	15	*p* = 0.653
Axillary Tail/UOQ	7	2	
UIQ	3	4	
LIQ	7	5	
LOQ	6	6	
Overlapping			
**Lymphovascular Invasion**	2	12	***p* < 0.001**
Yes	27	12	
No			

*p* < 0.05 considered significant (bolded). **OUQ**: upper outer quadrant, **UIQ**: upper inner quadrant, **LOQ**: lower outer quadrant, **LIQ**: lower inner quadrant.

**Table 3 diseases-11-00118-t003:** Univariate Logistic Regression of Nodal Positivity.

Covariate	Level	Number of Patients	Odds Ratio (95% CI)	*p* Value
**Age at diagnosis**	<60	42	5.0 (1.8–14.3)	**0.002**
	60+	31		
**T stage**	2–3	32	12.4 (4.1–37.6)	**<0.001**
	1mi or T1	41	-	
**Laterality**	Right	42	1.4 (0.6–3.7)	0.449
	Left	31	-	
**Surgery**	Mastectomy	37	11.8 (3.8–36.4)	**<0.001**
	Lumpectomy	36	-	
**Tumor location**	LIQ	7	1.6 (0.3–8.3)	0.576
	LOQ	12	0.9 (0.2–3.3)	0.821
	Overlapping	12	1.2 (0.3–4.5)	0.787
	UIQ	9	0.3 (0.1–1.9)	0.221
	Axillary Tail/OUQ	33	-	-
**Tumor size (mm)**		73	1.2 (1.1–1.3)	**<0.001**
**Tumor distance to nipple (mm)**		63	0.97 (0.95–0.99)	**0.009**
**Total number of suspicious nodes in image**		49	0.6 (0.3–1.4)	0.218

*p* < 0.05 considered significant (bolded). **OUQ**: upper outer quadrant, **UIQ**: upper inner quadrant, **LOQ**: lower outer quadrant, **LIQ**: lower inner quadrant.

**Table 4 diseases-11-00118-t004:** Multivariate Logistic Regression of Nodal Positivity.

Covariate	Level	Odds Ratio (95% CI)	*p*-Value
**Surgery Type**	Mastectomy	15.9 (3.7-68.5)	**<0.001**
	Lumpectomy	-	
**Tumor Size**	-	1.2 (1.1–1.3)	**<0.001**

Number of observations in the original data set = 73. Number of observations used = 73. *p* < 0.05 considered significant (bolded).

## Data Availability

No new data were generated or analyzed in this study. Data sharing is not applicable.

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
