# Peer review of "Tumor Characteristics Associated with Axillary Nodal Positivity in Triple Negative Breast Cancer"

_diseases, 2023, doi:10.3390/diseases11030118_

Round 1
Reviewer 1 Report (Previous Reviewer 1)
The strategic plan and graphical representation of data has been included.
Author Response
Point 1: The strategic plan and graphical representation of data has been included.
Response 1: Thank you for your previous suggestions. We believe that the strategic plan figure and graphical data representation have enhanced the readability of our manuscript.
Reviewer 2 Report (New Reviewer)
The Authors performed a really interesting study about Tumor Characteristics Associated with Axillary Nodal Positivity in Triple Negative Breast Cancer. It is in interesting topic. The paper is well written and interesting in all its field demonstrating a remarkable experience in the treatment of breast cancer.
In my opinion in patients undergoing surgery for breast cancer, there are several problems that should be better investigated. In order to better analyse this topic, I suggest considering the paper:
" Docimo G, Limongelli P, Conzo G, Gili S, Bosco A, Rizzuto A, Amoroso V, Marsico S, Leone N, Esposito A, Vitiello C, Fei L, Parmeggiani D, Docimo L. Axillary lymphadenectomy for breast cancer in elderly patients and fibrin glue. BMC Surg. 2013;13 Suppl 2(Suppl 2):S8. doi: 10.1186/1471-2482-13-S2-S8. Epub 2013 Oct 8. PMID: 24266959; PMCID: PMC3851152.."
“Parisi S, Ruggiero R, Gualtieri G, et al. Combined LOCalizer™ and Intraoperative Ultrasound Localization: First Experience in Localization of Non-palpable Breast Cancer. In Vivo. 2021;35(3):1669-1676. doi:10.21873/invivo.12426”
Author Response
Point 1: The Authors performed a really interesting study about Tumor Characteristics Associated with Axillary Nodal Positivity in Triple Negative Breast Cancer. It is in interesting topic. The paper is well written and interesting in all its field demonstrating a remarkable experience in the treatment of breast cancer. In my opinion in patients undergoing surgery for breast cancer, there are several problems that should be better investigated. In order to better analyse this topic, I suggest considering the paper:
" Docimo G, Limongelli P, Conzo G, Gili S, Bosco A, Rizzuto A, Amoroso V, Marsico S, Leone N, Esposito A, Vitiello C, Fei L, Parmeggiani D, Docimo L. Axillary lymphadenectomy for breast cancer in elderly patients and fibrin glue. BMC Surg. 2013;13 Suppl 2(Suppl 2):S8. doi: 10.1186/1471-2482-13-S2-S8. Epub 2013 Oct 8. PMID: 24266959; PMCID: PMC3851152.."
“Parisi S, Ruggiero R, Gualtieri G, et al. Combined LOCalizer™ and Intraoperative Ultrasound Localization: First Experience in Localization of Non-palpable Breast Cancer. In Vivo. 2021;35(3):1669-1676. doi:10.21873/invivo.12426
Response 1: Thank you for your kind remarks regarding our manuscript. We are delighted to learn that you found the paper's topic interesting and the content well-written. We appreciate you suggesting the two articles. The paper by Docimo et al., details the use of fibrin glue to reduce seroma complications after ALND. The study found that using fibrin glue reduces the magnitude and duration of seroma formation, but importantly does not prevent the occurance post-procedure. We have detailed these findings in the context of our study in the Introduction section paragraph 5. The paper by Parisi et al., discusses the use of wire-guided localization and intraoperative ultrasound for the detection and management of non-palpable breast cancer. We have referenced this study in the Discussion section paragraph 4, with regards to diseased lymph node detection.
Reviewer 3 Report (New Reviewer)
Dear respected authors,
I appreciate the valuable insights presented in your manuscript entitled "Tumor Characteristics Associated with Axillary Nodal Positivity in Triple Negative Breast Cancer". The article effectively addresses the research aims of evaluating factors associated with axillary lymph node involvement in TNBC and assessing the role of preoperative imaging. The introduction provides appropriate background (must be summaried), the methods section outlines the study's approach, the results section presents relevant findings, and the discussion section offers valuable interpretations. However, the article's quality could be improved with clearer formatting, such as separating the sections more distinctly and providing appropriate references for previous studies cited in the discussion.
Only a few minor comments:
Abstract: The abstract is a concise summary of your study. Consider revising the abstract to include more specific details about the findings, such as the exact odds ratios or effect sizes associated with larger tumor size and shorter tumor-to-nipple distance.
Introduction: Please make the background shorter and more to the point.
Methods: Provide more information about the imaging techniques used (ultrasound, MRI, mammogram). Include details about the protocol and procedures followed during imaging assessments. This will help readers understand the context and reliability of the imaging results.
Discussion: Expand on the clinical implications of your findings. How might these predictors (tumor size, distance to nipple) influence treatment decisions or patient management strategies? Discuss how these results can contribute to more individualized treatment approaches for TNBC patients.
Future directions: Consider adding a paragraph discussing potential future research directions. For instance, could additional studies be conducted to validate these findings in larger cohorts or across different healthcare settings? Are there implications for the development of targeted interventions or risk assessment models?
Author Response
Point 1: Abstract: The abstract is a concise summary of your study. Consider revising the abstract to include more specific details about the findings, such as the exact odds ratios or effect sizes associated with larger tumor size and shorter tumor-to-nipple distance.
Response 1: Thank you for your valuable feedback and thorough review of our manuscript. We have edited the abstract to inlcude more specific details and results.
Point 2: Introduction: Please make the background shorter and more to the point.
Response 2: The introduction has been significantly reduced and made to be more concise.
Point 3: Methods: Provide more information about the imaging techniques used (ultrasound, MRI, mammogram). Include details about the protocol and procedures followed during imaging assessments. This will help readers understand the context and reliability of the imaging results.
Response 3: We have included in the methods section extra information about imaging techniques and procuedures followed during assessment of imaging results.
Point 4: Discussion: Expand on the clinical implications of your findings. How might these predictors (tumor size, distance to nipple) influence treatment decisions or patient management strategies? Discuss how these results can contribute to more individualized treatment approaches for TNBC patients.
Response 4: We have added new information (highlighted in yellow) to the Discussion section to contextualize our findings within the currently available literature. Please also see section 5. Clinical Implications and Future Directions.
Point 5: Future directions: Consider adding a paragraph discussing potential future research directions. For instance, could additional studies be conducted to validate these findings in larger cohorts or across different healthcare settings? Are there implications for the development of targeted interventions or risk assessment models?
Response 5: Please see section 5. Clinical Implications and Future Directions.
Reviewer 4 Report (New Reviewer)
Triple negative breast carcinoma (TNBC) is a heterogeneous group of aggressive breast cancer, which currently is a focus of attention for new treatment and management. The study of lymph node status of TNBC and its correlation to clinical and pathological characteristics can assist in the management and improve the prognosis of the patients. Although this type of research has been done in breast cancer but has not been done specifically for TNBC. I have three comments for the manuscript:
-
1. In your study, only clinically lymph node negative patients are included for the study, therefore, strictly speaking, the clinical and pathologic characteristics associated with lymph node status and your conclusion in your study may to be true for those TNBC patients who are clinically lymph node positive. Perhaps this should be discussed in the discussion part and reflected in the title of the manuscript.
-
2. It is common assumption that lymphovascular invasion is associated with positive lymph node. In your manuscript, the pathologic data of the patients were also included for the data analysis. However, the lymphovascular invasion was not mentioned and analyzed. It will be interesting to know that in the clinically lymph node negative TNBC patient, lymphovascular invasion is associated with positive lymph node or not.
-
3. In some institutions, a fine needle aspirate or core needle biopsy of axillary lymph node will be performed for imaging abnormal axillary lymph node in preoperative breast cancer patient. In your study, the 43% positive axillary lymph node is quite high. Perhaps the role of axillary fine needle aspirate or core biopsy of the axillary lymph node should be mentioned in the discussion part.
Author Response
Point 1: In your study, only clinically lymph node negative patients are included for the study, therefore, strictly speaking, the clinical and pathologic characteristics associated with lymph node status and your conclusion in your study may not to be true for those TNBC patients who are clinically lymph node positive. Perhaps this should be discussed in the discussion part and reflected in the title of the manuscript.
Response 1: We sincerely appreciate your thoughtful evaluation of our manuscript. We believe that you raise an important point. We have edited the manuscript to emphasize that our cohort was fully clinically node negative. Please see highlighted regions of Results, Discussion and Conclusion where we discuss this finding further.
Point 2: It is common assumption that lymphovascular invasion is associated with positive lymph node. In your manuscript, the pathologic data of the patients were also included for the data analysis. However, the lymphovascular invasion was not mentioned and analyzed. It will be interesting to know that in the clinically lymph node negative TNBC patient, lymphovascular invasion is associated with positive lymph node or not.
Response 2: Please see Results section 3.3 and Table 2. We did indeed find that the presence of lymphovascular invasion was significantly associated with nodal positivity (p < 0.001).
Point 3: In some institutions, a fine needle aspirate or core needle biopsy of axillary lymph node will be performed for imaging abnormal axillary lymph node in preoperative breast cancer patient. In your study, the 43% positive axillary lymph node is quite high. Perhaps the role of axillary fine needle aspirate or core biopsy of the axillary lymph node should be mentioned in the discussion part.
Response 3: Please see Discussion section paragraph 4 where we have discussed role of US-FNA and core biopsy for lymph node detection. We have discussed these techniques in the context of the Z0011 study.
This manuscript is a resubmission of an earlier submission. The following is a list of the peer review reports and author responses from that submission.
Round 1
Reviewer 1 Report
Summary
Chintapally et al., have investigated the relationships between tumor size, location and distance to nipple relative to axillary node metastases in triple-negative breast cancer (TNBC) patients in the manuscript entitled, “Tumor Characteristics Associated with Axillary Nodal Positivity in Triple Negative Breast Cancer”. Authors show larger tumor size was associated with positive nodes. The tumor to nipple distance was shorter in node positive patients compared to node positive patients. The study concludes that larger tumor size and shorter tumor to nipple distance are independent predictors of nodal metastasis among TNBC patients. The study identified two key independent predictors of axillary lymph node metastasis in patients with TNBC. First being tumor size in mm and second being tumor to nipple distance. Authors observed a significant association between larger primary tumor size with higher rates of lymphovascular invasion.
Along with low sample size, the study shows a limitation where authors did not observe significant association between tumor location and axillary metastasis due to absence of TNBC tumors subsets located in the retro-areolar region. The current findings suggest that imaging modalities are not powerful enough as standalone tools in capturing lymph node positivity in TNBC patients. The findings from preoperative MRI, ultrasound, and mammograms may be pooled and used adjunctively to SLNB in TNBC. It is commendable that authors have recognized their limitation and have justified in the discussion.
Conceptual comments
The manuscript is clear, comprehensive and of relevance to the breast cancer field. The flow of the manuscript is smooth. Important articles in the recent past have been included and justified in the manuscript. There are no excessive self-citations. The data provided in the current manuscript is of great importance, relevance and a valuable addition to the existing knowledge of the scientific community.
Specific comments
It would be great to include figure in the manuscript describing the strategic plan of the study.
The demographic table 1 could be represented in graphical format (bar graph/ pie chart) for better data visualization.
If depicted in pictures/graphs, it would be easy to understand how the occurrence of primary tumors in different quadrants like UOQ/axillary tail are involved in axillary metastases.
Overall, the findings look interesting. However larger sample size would certainly help to gain better insights.
Reviewer 2 Report
In this paper submitted, the authors tried to evaluate the relationships between tumor size, location, and distance to nipple relative to axillary node metastases in triple-negative breast cancer (TNBC) patients, as well as the predictive capacity of imaging and showed that larger tumor size and shorter tumor to nipple distance are independent predictors of nodal metastasis among TNBC patients. This paper topic is interesting, well-written and the discussion provided regarding the results obtained is comprehensive and thus, it can be accepted for publication in the journal of Diseases.
Reviewer 3 Report
Dear Author (s)
1. There are a lot of grammatical errors. For example, you change "node positive" to "node-positive", "node negative" to "node-negative", "tumor to nipple" to "tumor-to-nipple", "biologic determinants" to "biological determinants", etc.
2. Some sentences need reference (s). For example, "Increased breast cancer screening, advancements in therapeutics, and other public health measures have contributed to the declining breast cancer mortality rates in the United States. However, breast cancer remains a significant health concern, with approximately 1 in 8 women having a chance of developing breast cancer in their lifetime.", etc.
3. Pay attention to the full name of each abbreviation for the first time. For example, in line 12, discussion, you should change "triple negative breast cancer" to "TNBC", etc.
4. Full name of ANKRD30A, ANP32E, DSC2, IL6ST, etc.
5. Please reduce the discussion.
6. Please add the full name of each abbreviation below each table or figure separately.
7. Please summarize the conclusion. You duplicate the results in the conclusion.
8. Please reduce the introduction.
9. Where are the references for "The TNBC subtype, continues to pose significant challenges in terms of diagnosis, prognosis, and treatment. Understanding the unique characteristics and factors associated with TNBC is crucial for improving patient outcomes and developing personalized treatment approaches. The evaluation of axillary lymph node involvement plays a pivotal role in staging and prognostic assessment, and further research is required to optimize the accuracy of preoperative imaging techniques in this regard".
10. Most sentences in the discussion have been duplicated with the results. For example, paragraphs 5, 6, and 7.
11. The analysis is very simple and it can say that the report is a simple epidemiology. You could add more analyses (Logistic Regression, graphs, etc.
There are a lot of grammatical errors. For example, you change "node positive" to "node-positive", "node negative" to "node-negative", "tumor to nipple" to "tumor-to-nipple", "biologic determinants" to "biological determinants", etc.
Reviewer 4 Report
Triple-negative breast cancer (TNBC) is an aggressive subtype of breast cancer where the lack of Estrogen, Progesterone, and Her2 receptors precludes it from the commonly available targeted drug treatments.
Chintapally et al. evaluated the relationship between tumor size, location, and distance to the nipple relative to axillary node metastases and the predictive capacity of imaging in TNBC from a retrospective cohort of 73 patients. Metastasis is the leading cause of morbidity and mortality in TNBC. They found that tumor size correlated with node positivity, tumor to nipple distance was shorter in node-positive patients and tumor quadrant was not associated with nodal metastasis.
This paper has major flaws that need to be addressed before publication.
The data in this paper are descriptive, with a few TNBC cases. It is hard to make clear conclusions. Axillary node status, tumor location, and nodal involvement are known predictors of outcomes in breast cancer patients. It is important to increase your sample size to improve validity and generalizability. It is also important to provide more information regarding case ascertainment and source population.
Minor points:
Tables formatting: improve readability by changing the layout and removing unnecessary data. For example, it is unclear from the current table format the distribution of age of this cohort or the race of the cohort.
Minor editing required